# Risk Stratification and Cancer Follow-Up: Towards More Personalized Post-Treatment Care in Canada

Robin Urquhart [1,2,*], Wendy Cordoba [1], Jackie Bender [3], Colleen Cuthbert [4,5], Julie Easley [6,7], Doris Howell [3], Julia Kaal [1,8], Cynthia Kendell [2], Samantha Radford [2] and Jonathan Sussman [9]

1.  Department of Community Health and Epidemiology, Faculty of Medicine, Dalhousie University, Room 413—Centre for Clinical Research, 5790 University Avenue, Halifax, NS B3H 1V7, Canada; wendy.cordoba@msvu.ca (W.C.); kjuliakaal@dal.ca (J.K.)
2.  Department of Surgery, Nova Scotia Health, Halifax, NS B3H 2Y9, Canada; cynthia.kendell@ccns.nshealth.ca (C.K.); samantha.radford@nshealth.ca (S.R.)
3.  Department of Supportive Care, Princess Margaret Cancer Research Institute, Toronto, ON M5G 0A3, Canada; jackie.bender@uhnresearch.ca (J.B.); doris.howell@uhn.ca (D.H.)
4.  Faculty of Nursing, University of Calgary, Calgary, AB T2N 4V8, Canada; cacuthbe@ucalgary.ca
5.  Cumming School of Medicine, University of Calgary, Calgary, AB T2N 4N1, Canada
6.  Department of Family Medicine, Dalhousie University, Halifax, NS B3J 3T4, Canada; julie.easley@horizonnb.ca
7.  Department of Medical Education, Horizon Health Network, Fredericton, NB E3B 4R3, Canada
8.  Research, Innovation, and Discovery, Nova Scotia Health, Halifax, NS B3H 1V7, Canada
9.  Department of Oncology, McMaster University, Hamilton, ON L8V 5C2, Canada; sussman@hhsc.ca
*   Correspondence: robin.urquhart@nshealth.ca; Tel.: +902-473-7290; Fax: +902-473-4631

**Abstract:** After treatment, cancer survivors require ongoing, comprehensive care to improve quality of life, reduce disability, limit complications, and restore function. In Canada and internationally, follow-up care continues to be delivered most often by oncologists in institution-based settings. There is extensive evidence to demonstrate that this model of care does not work well for many survivors or our cancer systems. Randomized controlled trials have clearly demonstrated that alternate approaches to follow-up care are equivalent to oncologist-led follow-up in terms of patient outcomes, such as recurrence, survival, and quality of life in a number of common cancers. In this paper, we discuss the state of follow-up care for survivors of prevalent cancers and the need for more personalized models of follow-up. Indeed, there is no one-size-fits-all solution to post-treatment follow-up care, and more personalized approaches to follow-up that are based on individual risks and needs after cancer treatment are warranted. Canada lags behind when it comes to personalizing follow-up care for cancer survivors. There are many reasons for this, including difficulty in determining who is best served by different follow-up pathways, a paucity of evidence-informed self-management education and supports for most survivors, poorly developed IT solutions and systems, and uneven coordination of care. Using implementation science theories, approaches, and methods may help in addressing these challenges and delineating what might work best in particular settings and circumstances.

**Keywords:** survivorship; follow-up care; personalized care

## 1. Introduction

Based on current projections, one in every two Canadians will develop cancer in their lifetime and the number of cancer cases will be 79% higher in 2028–2032 than in 2003–2007 [1]. As the number of Canadians diagnosed with cancer increases, the likelihood that a person will survive cancer has also increased. In fact, the prevalence of cancer survivors has risen dramatically over the past two decades: 63% of all people diagnosed in Canada today will live for 5 years or more after their diagnosis [1]. Among the most common cancers, the 5-year net survival rate is currently 93% for prostate cancer, 88% for breast cancer, and 65% for colorectal cancer [1]. These outcomes reflect tremendous

success in early detection and cancer treatments. Despite enhanced survival, a diagnosis of cancer and its treatment have substantial late and long-term adverse impacts. In fact, the magnitude of survivors' medical and supportive care needs after treatment is similar to the magnitude of needs they experience during treatment [2]. Moreover, many survivors are at risk of developing late effects from their treatment, including a second cancer, osteoporosis, and organ dysfunction (e.g., heart and lung disease), that may not present until years after completion of treatment. Consequently, survivors require ongoing, comprehensive care to improve quality of life, reduce disability, limit complications, and restore function. Therefore, follow-up care after cancer treatment is now recognized as a component of high-quality medical care. According to the Institute of Medicine in the United States, optimal follow-up care involves: (1) Surveillance for recurrence and new cancers; (2) prevention and management of the long-term and late effects of both the cancer and its treatment; and (3) promotion of healthy behaviors to mitigate new and ongoing health concerns [3].

For many prevalent cancer types, evidence exists to guide optimal follow-up care. Follow-up care guidelines are available in many jurisdictions to guide the timing of physician visits and surveillance investigations, as well as strategies to address late and long-term effects and support changes in lifestyle behaviors. This guidance is disease-specific given that different cancers will necessitate different therapies and have different risk profiles when it comes to recurrence and late effects. Moreover, many organizations provide tools, such as follow-up care pathways and survivorship care plans to support quality follow-up, as well as discharge letter templates to transfer critical information during transitions in care. For example, Cancer Care Ontario has developed and implemented pathway maps for many cancers (e.g., skin, colorectal, prostate, breast) that help the treating physician (e.g., oncologist) in managing follow-up care and determining whether and/or at what timepoint a patient can be discharged to primary care. Furthermore, the organization's 2019 guideline on models of follow-up care provides recommendations around a number of supports that should be in place (e.g., follow-up care plans, self-management programs) to facilitate transition to primary care and ensure timely re-entry into the cancer system if/when needed [4].

In this paper, we focus on the follow-up care of prevalent cancers with high survival. Clearly, the nature of follow-up care will differ depending on the cancer itself, its treatment, and the risk of long-term and late effects.

## 2. Models of Follow-Up Care

In Canada and internationally, follow-up care has most often been delivered by oncologists in institution-based settings (i.e., hospital-based outpatient oncology clinics). However, this model has not worked well for patients or our cancer systems. For example, research across Canada has consistently found that survivors feel unprepared for the follow-up care period, including the ongoing psychological and emotional burden they experience after the treatment, report a lack of access to timely information and supports during follow-up, experience poor coordination across sectors, and have many chronic needs that remain unmet [5–9]. Concurrently, cancer systems in Canada have been challenged for some time now with limited resources—including lack of time and staff—to provide optimal care for both newly diagnosed cancer patients and cancer survivors [10–13]. In fact, the demand for oncology services may be quickly exceeding the supply. A US study by the American Society of Clinical Oncology estimated that demand for oncology services would rise by 48% between 2005 and 2020, whereas the supply of services during the same time would grow more slowly at approximately 14% [14].

A solution has been provided through research: For 25 years, randomized controlled trials (RCTs) have shown that alternate approaches to follow-up care, including primary care provider-led follow-up, are equivalent to oncologist-led follow-up in terms of patient outcomes, such as recurrence, survival, and quality of life [15–25]. When there are differences, the outcomes (e.g., patient satisfaction, adherence to guideline recommendations, patient and societal costs) typically favor the alternate approaches [18,20–24]. Consistent

with these RCT findings, a 2014 review of best practices for follow-up care of breast, colorectal, prostate, and lung cancers found no differences in recurrence, survival, and patient well-being between shared care follow-up (oncologist/primary care) and primary care-led follow-up care [26].

These long-standing RCT findings have been confirmed by real-world evidence from Ontario. Following the publication of its 2012 guideline Models of Care for Cancer Survivorship [27], the provincial cancer agency, Cancer Care Ontario, initiated a demonstration project to provide support for regional cancer programs to develop models of care that included transition to primary care for low-risk breast and colorectal cancer survivors. Under this project, survivors were eligible for transition if they completed their cancer treatment, had no evidence of disease or ongoing cancer-related issues, and were deemed appropriate for community-based care. Between 2012–2015, more than 10,000 breast and colorectal cancer survivors were transitioned through these new models [4]. Subsequent evaluation revealed that clinical and quality outcomes were equivalent in low-risk survivors who were transitioned back to primary care after treatment compared with propensity-matched survivors who did not experience transition [28]. Moreover, transitioned survivors had greater rates of surveillance mammography (recommended care), fewer other diagnostic tests (non-recommended care), and incurred lower costs to the health system. Taken together, the body of evidence suggests that, for many cancer survivors, primary care-led follow-up care is highly appropriate. Nevertheless, there is wide variation across Canada in terms of models of follow-up, with many cancer survivors continuing to see their oncology teams in the long-term [29–32].

### 3. Towards More Personalized Follow-Up Care

Given the recognition of no one-size-fits-all solution, there have been growing calls from the cancer survivorship community to develop more personalized approaches to follow-up care that are based on individual risks and needs after cancer treatment [33–36]. Supporters of 'stratified' follow-up care argue that survivors with low needs or who are at low risk of adverse outcomes are best served by transition to primary care or even self-managed follow-up, and that those with higher needs and/or at higher risk are best served by some level of involvement of their oncology teams. The National Health Service in the United Kingdom (UK) is leading in this regard, and has implemented personalized stratified follow-up pathways to tailor follow-up care to individual needs and conditions. Colorectal, breast, and prostate cancer survivors are triaged to one of three different follow-up care pathways based on: The severity of their ongoing treatment effects; risk of recurrence, subsequent cancers, and late effects; functional ability; psychological health; and social circumstances [34,37]. The three pathways are: (1) Supported self-management, with remote monitoring of surveillance tests and results; (2) a coordinated care approach (which may be primary care- or nurse-led); or (3) complex care management, in which patients with high needs are treated by a multidisciplinary team of oncology providers. Pathway selection is a shared decision between patients and providers, and patients can move to a new pathway as needs change. Key elements of these pathways include remote monitoring, agreed-upon systems for rapid re-access, needs assessments and care planning, treatment summaries, self-management education, and access to support services, such as psychological, diet, and physical activity counselling [38].

Evaluation of the stratified pathways in England demonstrated that the self-managed pathway (followed by more than half of survivors) meets survivors' needs, frees specialists' time, and increases efficiency of the system, with an estimated savings of GBP 90 million over 5 years [39,40]. The same pathways were subsequently tested in Northern Ireland for breast cancer patients, with 58% of all breast cancer survivors selecting the self-management pathway. Health system outcomes demonstrated a freeing up of clinic visits in surgery and oncology that provided clinicians more time to spend with patients who had complex needs, improvements in the receipt of timely follow-up mammograms (recommended care) by 20%, and a decrease in wait times for surgery and medical oncology by 34% [41]. Both

Australia and the US are reforming follow-up care using the same principles as those used in the UK models.

### 4. Barriers to Operationalizing Personalized Follow-Up Care in Canada

In many ways, Canada has been an international leader in cancer survivorship research, including the testing of models of follow-up care, as well as personalized cancer care. Despite this work, Canada has fallen behind other jurisdictions in the implementation and testing of personalized follow-up care for cancer survivors. There may be many reasons for this, including difficulty determining who is best served by different follow-up pathways, a paucity of evidence-informed self-management education and supports for most survivors, poorly developed IT solutions and systems, and uneven coordination of care.

First, there are gaps in our understanding of how to identify those best suited by particular follow-up pathways. For example, what makes a person low, medium or high risk? In the UK, criteria do exist to help in identifying the best pathway for any given patient. However, these criteria do not follow complex algorithms. In fact, pilot testing demonstrated that a complex algorithm was unnecessary [33]. Rather, at its core, risk stratification required simply identifying patients who need close follow-up, and then supporting the majority of patients to self-manage their health and recovery with limited clinician involvement, except for surveillance or screening tests and investigations. Nevertheless, clinician agreement on what the pathways 'look like' and the specific criteria that would lead an individual down a specific pathway will be key to integrating stratified pathways into routine care. Agreement on a concept ('stratified follow-up care') that is ill-defined and likely unfamiliar to many providers and policymakers will be a challenge. A deliberative consultation on stratified follow-up pathways in Quebec found that the concept of risk stratification was ambiguous and that the lack of a clear definition of 'risk' was problematic [42]. In a similar vein, not all patients will accept care that is provided by non-oncologists. The use of alternative approaches to follow-up care must be a shared decision and will not work for everyone given their diagnosis, therapies, and nature of long-term effects. Indeed, some patients will require ongoing oncologist follow-up due to their risk of recurrence and/or severity or complexity of ongoing effects. However, the introduction of alternative follow-up pathways to survivors at 'low risk' will ultimately require that oncologists themselves trust in these alternative pathways.

Second, self-management education and supports are a key component of stratified follow-up pathways [43]. Indeed, self-management support programs reduce physical and emotional distress and improve quality of life in cancer patients and survivors [44–46]. However, cancer systems lag behind other chronic disease systems in implementing self-management as a routine part of care [47], and few cancer centers have integrated comprehensive self-management programs in Canada. Although work has occurred to develop online self-management programs in the country [48] and self-management in cancer has become a quality standard [49], considerable work has to be conducted to ensure cancer patients and survivors benefit from self-management programming more broadly and in a more sustainable way [47]. Even when resources do exist, many studies demonstrate that Canadian cancer survivors are largely unaware of the resources available [50–52]. In addition, rural and Indigenous survivors report additional barriers to accessing supportive care services in Canada [53–55]. These latter points underscore the need for integrated self-management education and supports.

Third, a key element to support the systematic use of stratified follow-up pathways work in the UK is the development of automated IT systems to enable remote monitoring and reporting of results for individuals who select the supported self-managed pathway. In Canada, the use of automated systems to schedule routine (surveillance) tests and investigations, and electronically report results to both cancer patients and their primary care providers, is not widespread (if it exists at all). Although innovations in virtual care and remote monitoring and management of cancer patients' symptoms have occurred during the global COVID-19 pandemic, technological solutions that allow oncology providers to

remotely schedule, monitor, and report follow-up tests have not been implemented. In the UK, supported self-management was increasingly viewed as a safe and acceptable follow-up pathway as providers' confidence and trust in remote monitoring solutions increased [40].

A fourth challenge to operationalizing personalized follow-up care in Canada relates to the overall lack of coordination and communication between cancer patients and providers, and between providers. This is a long-standing and well recognized challenge in the care of people with cancer [12,56]. Improving care—and indeed, personalizing care—will require a coordinated approach across providers, and good communication between primary care and oncology teams. As described by Tremblay et al. [42], risk stratification requires coordinated care approaches, and coordinated models necessitate information sharing and a clear delineation of responsibilities, roles, and tasks. Participants in their deliberative sessions pointed to the lack of coordination between oncology and primary care, and even the limited communication and coordination amongst members of oncology teams, as barriers to implementing risk stratified follow-up care.

## 5. Future Directions

While the above noted gaps are certainly not the only ones, they need to be addressed to successfully implement personalized follow-up care in consistent and sustainable ways. Decades of research have shown that alternative models of care for cancer survivors are equivalent to or better than oncologist-led care. Nevertheless, follow-up care after cancer treatment continues to be haphazard in Canada, and certainly not optimal for cancer survivors, providers or health systems. The challenge now is to establish how to implement risk- or needs-based approaches to follow-up care within our existing systems. A clear place to start is to understand both survivors' and providers' acceptability of these approaches, and to achieve consensus on what is feasible given local resources and constraints. In the UK, senior leadership commitment, broad stakeholder engagement, and clinician agreement on the criteria for each pathway were key supports to implementing stratified follow-up care [41]. Situating future work in implementation science [57,58] and systems thinking [59,60] will be key to identifying, planning, evaluating, and learning from initiatives that begin to unravel this 'wicked problem' in cancer care [42]. As a first step, this should mean involving team members with implementation science expertise into both research and clinical care endeavors.

From a cancer system perspective, there is clearly a need across Canada to increase capacity in self-management education and supports. This will necessitate providers who are equipped in providing this type of education and training. Moreover, researchers have noted the need for implementation science in the area of self-management [47,61] to understand core components of self-management interventions (e.g., the essential ingredients that lead to consistent and sustained patient outcomes) and the determinants of embedding self-management supports into complex care environments.

From a technology standpoint, researchers and clinical teams should look to the innovations that have emerged during the global COVID-19 pandemic for virtual care as potential solutions that may be leveraged and/or adopted to enable remote monitoring in survivorship care. Furthermore, providers' and patients' increased comfort with virtual care as a result of the pandemic represents an opportunity to advance IT solutions that may not have been present prior to March 2020. Post-pandemic, virtual models of follow-up care may also be a long-term alternative to in-person, specialist-led follow-up care. Although limited research exists, studies have demonstrated that virtual care (predominantly telemedicine) appears to not compromise patient satisfaction and safety [62] and may be acceptable to survivors with low ongoing needs [63]. Interestingly, studies have shown that survivors of pediatric and adolescent cancers are highly satisfied with virtual long-term follow-up [64–67]. A large evaluation of the shift to virtual cancer care in the province of Alberta, Canada, during the first wave of the COVID-19 pandemic [68] found that virtual care worked best for patients with few symptoms and good health status, and those

who lived in rural or remote areas. However, an important finding was the staff's lack of confidence in addressing patient needs in a virtual care setting. The acceptability and effectiveness of virtual follow-up care, including what works, for whom, and under what circumstances, warrants ongoing research.

To date, a key gap in the existing research pertains to understanding the acceptability and effectiveness of alternative models of care across diverse population groups [69]. Understanding whether and how personalized approaches to follow-up can address disparities in care and supports will be key to ensuring that all survivors receive the care they need to optimize quality of life, reduce disability, limit complications, and restore function. Similarly, tailoring follow-up pathways that account for the unique circumstances and preferences of certain groups, such as pediatric and adolescent cancer survivors, will be critical to maximize recovery and outcomes.

Finally, any new initiatives must be evaluated to ensure they benefit patients and our health systems. Real-world evidence around changes in models of follow-up care, such as the work in Ontario related to transition to primary care (described above) [28], would offer important information to decision makers on key dimensions of health care quality, including safety, effectiveness, equity, and efficiency. Where possible, evaluations should include an assessment of the cost-effectiveness of new models, as well as incorporate outcomes that are especially important to survivors post-treatment [70].

## 6. Conclusions

The current models of post-treatment follow-up care do not work well for many cancer survivors or cancer systems. In short, we must do better in Canada to meet cancer survivors' needs and support their health recovery, as well as meet the challenges faced by our cancer systems. As our research and clinical communities embrace more personalized approaches to cancer care, post-treatment follow-up care should be a part of that conversation. Risk stratified follow-up care, where one's model of follow-up is personalized based on individual risks, needs, and/or circumstances, has been promoted for more than a decade, yet its implementation into practice has been slow to non-existent in Canada. There are real challenges to operationalizing stratified pathways into routine care. Using implementation science theories, approaches, and methods may help in addressing these challenges and delineating what might work best in particular settings and circumstances.

**Author Contributions:** Conceptualization, all authors; writing—original draft preparation, R.U.; writing—review and editing, all authors; funding acquisition, R.U. All authors have agreed to be personally accountable for the author's own contributions. All authors have read and agreed to the published version of the manuscript.

**Funding:** This research was funded by a subgrant from the Canadian Team to Improve Community-Based Cancer Care along the Continuum (CanIMPACT), which is funded by a Team Grant from the Canadian Institutes of Health Research (TT7-128272).

**Institutional Review Board Statement:** This commentary did not require Institutional Review Board approval.

**Informed Consent Statement:** Not applicable.

**Data Availability Statement:** Not applicable.

**Conflicts of Interest:** The authors declare no conflict of interest. The funder had no role in the design, execution, interpretation, or writing of the paper.

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
