# Peer review of "Risk Stratification and Cancer Follow-Up: Towards More Personalized Post-Treatment Care in Canada"

_curroncol, doi:10.3390/curroncol29050261_

Round 1

Reviewer 1 Report

The authors of this paper have provided commentary regarding current practices for follow-up care among cancer survivors in Canada, with international evidence to support the use of risk stratified pathways. The authors present a well-rounded case for moving towards a risk-stratified model of follow-up care including challenges and future directions to progress this field.

Author Response

Comment: The authors of this paper have provided commentary regarding current practices for follow-up care among cancer survivors in Canada, with international evidence to support the use of risk stratified pathways. The authors present a well-rounded case for moving towards a risk-stratified model of follow-up care including challenges and future directions to progress this field.

Response: Thank you.

Reviewer 2 Report

This manuscript is about commentary on the follow-up system for cancer patients, the challenges, advantages, and disadvantages.

the actual form of the article can be accepted as a general speech article, how to advise the medical centers to establish their follow-up systems and improve them regarding the requirements, pathways, and timing how to do that.

Author Response

Comment: This manuscript is about commentary on the follow-up system for cancer patients, the challenges, advantages, and disadvantages. The actual form of the article can be accepted as a general speech article, how to advise the medical centers to establish their follow-up systems and improve them regarding the requirements, pathways, and timing how to do that.

Response: Thank you.

Reviewer 3 Report

The abstract is incomplete, inadequate, and lacks structure and substance. As a matter of fact, the abstract looks like a snapshot of a review of the literature with no purpose or objective stated.  

The Introduction is incomplete. The introduction should be expanded to include a narrative on the importance of follow-up care guidelines, follow-up care plans, and follow-up care schedules, underscoring the fact that cancer is not a single diagnosis with a single treatment.

Providing examples of follow-up care for the most common cancers would provide a better understanding of the cancer-specific nature of follow-up care.

The narrative of the commentary should be reconceptualized to present alternative follow-up care models as additional task-shifting options to existing practices and not as substitutes for established practices.

A statement on how to address cancer care follow-up disparities as well as how cancer care models could be tailored across pediatric, adult, and older cancer survivors would be useful.

More could have been stated about digital care follow-up and the perspective of patient-centered cancer care, including cost implications of long-term cancer care and approaches to cost-effective care while maintaining the quality of care.

The challenges and limitations of alternative follow-up models should be clearly articulated. Not all cancer patients may accept the care provided by non-oncologists.

Author Response

Comment: The abstract is incomplete, inadequate, and lacks structure and substance. As a matter of fact, the abstract looks like a snapshot of a review of the literature with no purpose or objective stated.

Response: Thank you for this comment. We followed the journal guidelines for the abstract, which specifically indicates no headings for the abstract (unstructured). We have also reviewed the abstracts of recently published commentaries to determine how we can improve the abstract. We have provided a purpose statement in the abstract of the revised manuscript but otherwise we are uncertain what specifically to do to improve it. Additional guidance from the Reviewer or Editor is welcome.

Comment: The Introduction is incomplete. The introduction should be expanded to include a narrative on the importance of follow-up care guidelines, follow-up care plans, and follow-up care schedules, underscoring the fact that cancer is not a single diagnosis with a single treatment.

Response: The Reviewer is indeed correct in that the introduction section is not a comprehensive overview of follow-up care. We had attempted to provide much of the content of the commentary, including context around the importance of follow-up care, in subsequent sections. However, we have added the following paragraph to the Introduction of the revised manuscript to provide more context around follow-up care and the tools that exist to support quality follow-up:

For many prevalent cancer types, evidence exist to guide optimal follow-up care. Follow-up care guidelines are available in many jurisdictions to guide the timing of physician visits and surveillance investigations, and strategies to address late and long-term effects and support changes in lifestyle behaviors. This guidance is disease-specific given that different cancers will necessitate different therapies and have different risk profiles when it comes to recurrence and late effects. Many organizations also provide tools such as follow-up care pathways and survivorship care plans to support quality follow-up, as well as discharge letter templates to transfer critical information during transitions in care. Cancer Care Ontario, for example, has developed and implemented pathway maps for many cancers (e.g., skin, colorectal, prostate, breast) that help the treating physician (e.g., oncologist) manage follow-up care and determine whether and/or at what timepoint a patient can be discharged to primary care. The organization’s 2019 guideline on models of follow-up care also provides recommendations around a number of supports that should be in place (e.g., follow-up care plans, self-management programs) to facilitate transition to primary care and ensure timely re-entry into the cancer system if/when needed.4

Comment: Providing examples of follow-up care for the most common cancers would provide a better understanding of the cancer-specific nature of follow-up care.  [This commentary purposely focused on prevalent cancers with high survival … we entirely acknowledge that the nature of follow-up care will differ depending on the cancer itself …]

Response: Thank you for this comment. Please see the paragraph added above. We have also included the following statement at the end of the Introduction section:

In this paper, we focus on the follow-up care of prevalent cancers with high survival. Clearly, the nature of follow-up care will differ depending on the cancer itself, its treatment, and the risk of long-term and late effects.

Comment: The narrative of the commentary should be reconceptualized to present alternative follow-up care models as additional task-shifting options to existing practices and not as substitutes for established practices.

Response: We respectfully disagree with this comment. In this commentary, we are in fact suggesting that existing models do not work well for many survivors (or cancer systems) and that alternative models of follow-up are needed (for many survivors) to replace existing practices. The former point is backed by decades of research in Canada and beyond. This certainly does not mean that there isn’t a subgroup of survivors who would benefit from the traditional model – this is indeed the case and one of the pathways in risk-stratified follow-up care. Yet, it is our premise that this is, based on the evidence, a minority of cancer survivors from the prevalent cancer groups with high survival (e.g., breast, colorectal, and prostate cancers). Given that this is a commentary wherein we are laying out an argument based on evidence, we do not believe this paper should be reconceptualized. We do, however, welcome additional commentary that advances alternative perspectives.

Comment: A statement on how to address cancer care follow-up disparities as well as how cancer care models could be tailored across pediatric, adult, and older cancer survivors would be useful.

Response: This is a great comment. Thank you. We have added the following statement(s) to the revised manuscript for Future Directions:

Understanding whether and how personalized approaches to follow-up can address disparities in care and supports will be key to ensuring that all survivors receive the care they need to optimize quality of life, reduce disability, limit complications, and restore function. Similarly, tailoring follow-up pathways that account for the unique circumstances and preferences of certain groups, such as pediatric and adolescent cancer survivors, will be critical to maximize recovery and outcomes.

Comment: More could have been stated about digital care follow-up and the perspective of patient-centered cancer care, including cost implications of long-term cancer care and approaches to cost-effective care while maintaining the quality of care.

Response: Thank you for this comment. We have now incorporated the following statements in the revised manuscript:

Post-pandemic, virtual models of follow-up care may also become a long-term alternative to in-person, specialist-led follow-up care. Although limited research exists, studies have demonstrated that virtual care (predominantly telemedicine) appears not to compromise patient satisfaction and safety62 and may be acceptable to survivors with low ongoing needs.63 Interestingly, studies have shown that survivors of pediatric and adolescent cancers are highly satisfied with virtual long-term follow-up.64-67 A large evaluation of the shift to virtual cancer care in the province of Alberta, Canada, during the first wave of the COVID-19 pandemic68found that virtual care worked best for patients with few symptoms and good health status, and those who lived in rural or remote areas. An important finding, however, was staff’s lack of confidence in addressing patient needs in a virtual care setting. The acceptability and effectiveness of virtual follow-up care, including what works, for whom, and under what circumstances, warrants ongoing research.

Where possible, evaluations should include an assessment of the cost-effectiveness of new models as well as incorporate outcomes that are especially important to survivors post-treatment.69

Comment: The challenges and limitations of alternative follow-up models should be clearly articulated. Not all cancer patients may accept the care provided by non-oncologists.

Response: Absolutely, we are in agreement on this point. Not all cancer patients will accept care that is provided by non-oncologists. In this paper, we are advocating for a personalized approach that is best suited to any particular patient. In the UK, the specific pathway a person goes down is a shared decision between the patient and their care team. Within a personalized approach, we expect that 1) some proportion of patients will indeed wish to stay with their oncologists and 2) some proportion of patients should stay with their oncologists given the nature of their disease, ongoing therapies, and nature of their long-term effects. We have added the following statement to the revised manuscript:

In a similar vein, not all patients will accept care that is provided by non-oncologists. The use of alternative approaches to follow-up care must be a shared decision and will not work for everyone given their diagnosis, therapies, and nature of long-term effects. However, the introduction of alternative follow-up pathways to survivors at ‘low risk’ will ultimately require that oncologists themselves trust in these alternative pathways.

Round 2

Reviewer 3 Report

This reviewer will focus on the response by the authors to one of the comments made in a previous review as follows.  The narrative of the commentary should be reconceptualized to present alternative follow-up care models as additional task-shifting options to existing practices and not as substitutes for established practices.” While the authors agree with this reviewer that some cancer patients may not accept therapy offered by non-oncologists, they have not addressed this important comment properly: This review believes that sound commentary based on the prevailing scientific evidence is more helpful to patients than hortatory viewpoints designed to drive home unsubstantiated preconceived ideas. Please read the evidence provided in the reference below on why alternative models CAN NOT YET REPLACE existing models and address this reviewer's comment appropriately:

Chan, R. J., Crawford-Williams, F., Crichton, M., Joseph, R., Hart, N. H., Milley, K., ... & Nekhlyudov, L. (2021). Effectiveness and implementation of models of cancer survivorship care: an overview of systematic reviews. Journal of Cancer Survivorship, 1-25.

Author Response

With respect, we are uncertain what the Reviewer wants us to do in relation to this comment. We are also uncertain what exactly the Reviewer means by stating we have “hortatory viewpoints designed to drive home unsubstantiated preconceived ideas.” We are familiar with the evidence on alternative models of survivorship care and have considered it carefully in the preparation of this commentary. In fact, the Chan et al 2021 paper, suggested by the Reviewer, is very much aligned with our commentary.

Nowhere in the commentary do we state that existing models need to be replaced. In our initial response to this comment, we did use that language (in the response only, not the paper) and perhaps we should have been more nuanced in our language. In our commentary, we are emphasizing a more personalized approach to follow-up care “based on individual risks, needs, and/or circumstances” (as written in our commentary). This is what we intended by the word “replace” in our initial response – we should “replace” the notion that there is a one-size-fits-all approach with the notion that we need systems to provide more personalized care. Some survivors will absolutely still require ongoing oncologist led-care due to their risk of recurrence and/or severity of ongoing problems. Any movement toward personalized model of care will include this as a possible follow-up approach/pathway.

The Chan et al 2021 paper is an excellent paper that lays out the evidence around models of follow-up care. Our commentary is not intended to provide such an in-depth examination of the evidence. The evidence we do cite, however, is aligned with the evidence provided in the Chan et al paper. For example, we stated that “for 25 years, randomized controlled trials (RCTs) have shown that alternate approaches to follow-up care, including primary care provider-led follow-up, are equivalent to oncologist-led follow-up in terms of patient outcomes such as recurrence, survival, and quality of life.15-25 When there are differences, the outcomes (e.g., patient satisfaction, adherence to guideline recommendations, patient and societal costs) typically favor the alternate approaches.18,20-24” Moreover, the very last sentence of the Chan et al paper states: “Rather than aiming for an optimal “one-size fits all” model of survivorship care, care needs to be tailored to the individual, with appropriate models resulting in improved outcomes and healthcare efficiency.” This is indeed the premise of our commentary.

Having said these things, and being unclear about what precisely the Reviewer is hoping us to change, we have added the following two statements to the revised version of the commentary to be clear that oncologist-led care will still be required for some patients:

Indeed, some patients will require ongoing oncologist follow-up due to their risk of recurrence and/or severity or complexity of ongoing effects.

To date, a key gap in the existing research pertains to understanding the acceptability and effectiveness of alternative models of care across diverse population groups.69